# Bismuth−Antimony Alloy Embedded in Carbon Matrix for Ultra-Stable Sodium Storage

**DOI:** 10.3390/ma16062189

**Published:** 2023-03-09

**Authors:** Wensheng Ma, Bin Yu, Fuquan Tan, Hui Gao, Zhonghua Zhang

**Affiliations:** Key Laboratory for Liquid-Solid Structural Evolution and Processing of Materials (Ministry of Education), School of Materials Science and Engineering, Shandong University, Jingshi Road 17923, Jinan 250061, China

**Keywords:** bismuth-antimony anodes, carbon framework, pyrolysis method, XRD, sodium-ion batteries

## Abstract

Alloy-type anodes are the most promising candidates for sodium-ion batteries (SIBs) due to their impressive Na storage capacity and suitable voltage platform. However, the implementation of alloy-type anodes is significantly hindered by their huge volume expansion during the alloying/dealloying processes, which leads to their pulverization and detachment from current collectors for active materials and the unsatisfactory cycling performance. In this work, bimetallic Bi−Sb solid solutions in a porous carbon matrix are synthesized by a pyrolysis method as anode material for SIBs. Adjustable alloy composition, the introduction of porous carbon matrix, and nanosized bimetallic particles effectively suppress the volume change during cycling and accelerate the electrons/ions transport kinetics. The optimized Bi_1_Sb_1_@C electrode exhibits an excellent electrochemical performance with an ultralong cycle life (167.2 mAh g^−1^ at 1 A g^−1^ over 8000 cycles). In situ X-ray diffraction investigation is conducted to reveal the reversible and synchronous sodium storage pathway of the Bi_1_Sb_1_@C electrode: (Bi,Sb) 
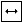
 Na(Bi,Sb) 
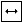
 Na_3_(Bi,Sb). Furthermore, online electrochemical mass spectrometry unveils the evolution of gas products of the Bi_1_Sb_1_@C electrode during the cell operation.

## 1. Introduction

Lithium-ion batteries (LIBs), the leader in portable electrochemical energy storage devices, are not suitable for large-scale applications due to their high processing cost and limited lithium resources [1]. Sodium-ion batteries (SIBs) are appealing due to their price advantage and resource richness. At present, the key issue in promoting SIBs is to find suitable anodes with an excellent cycling lifespan and a high capacity. Alloying-type anodes (Bi, Sb, P, and Sn etc.) have become promising candidates for SIBs for their high theoretical specific capacities compared with carbon-based materials and their suitable voltage platform [2]. However, the fatal defect of huge volume changes occurring during cycling makes the alloying-type anode impracticable. The large volume changes lead to pulverization and detachment from the current collector of the active materials, resulting in rapid capacity fading. In addition, the sluggish diffusion of Na^+^ ions into alloying anodes is another obstacle to the high-power density of SIBs [3].

To address these issues, researchers often adopt the following strategies: (1) the anodes with unique nanostructures, such as nanoflowers [4], nanoarrays [5], nanotubes [6], nanorods [7], etc. are designed to alleviate the repetitive volume stress during cycling and facilitate the transportation of electron and ions. (2) Bimetallic alloys are constructed, including inactive/active and active/active types, in which another metal serves as a buffer matrix, effectively enhancing structural stability. (3) The classical design of the metal/carbon nanocomposite acts as an achievable strategy for enhancing Na storage performance. Sb-C and Bi-C composite materials (Bi@C [8,9], Sb@C [10], and Sb/rGO-C [11], etc.) have been proposed as high-performance SIBs anodes. The combination of carbon coating and alloying strategy outperforms the single approach for improving electrochemical performance. For example, Wang et al. reported that CoSb nanocrystallites in a honeycomb-like carbon matrix (CoSb@3DPCs) demonstrated remarkable cycling durability up to 500 cycles and accelerated electron/ion transformation for a high-rate performance [12].

Bi and Sb atoms can form a series solid solution with continuous components; therefore, Bi−Sb alloys have become an important research topic for anodes of SIBs due to their unique physicochemical properties [13,14,15]. In Bi−Sb alloys, the high capacity of antimony (660 mAh g^−1^) and the relatively low volume expansion of bismuth (250%) can be combined to achieve outstanding electrochemical performance (high specific capacity, rate capability, and long cycling lifespan) [2]. Guo et al. reported that the lattice-softening mechanism in Bi−Sb alloy nanoparticles can improve cycling durability for SIBs. Bi−Sb alloys have a lower elastic modulus and higher toughness than single metals, demonstrating their excellent anti-pulverization capability and durability for Na^+^ insertion and extraction [16]. Therefore, the preparation of carbon-coated Bi−Sb alloy anodes is a feasible way for high-performance anodes of SIBs. However, problems such as complicated fabrication and high costs still hinder the commercialization of alloying anodes. It remains a key challenge to prepare metal/carbon nanocomposites with facile and scalable methods.

Herein, taking advantage of the capture effect of the porous carbon matrix, the Bi−Sb nanocomposites with tunable alloy compositions were successfully fabricated by one-step pyrolysis of bismuth (III) citrate and potassium antimony tartrate precursors. The continuous composition of the Bi−Sb solid solution can be obtained by adjusting the feed ratio of the precursor. The porous carbon framework generated by pyrolysis restricts the aggregation of Bi−Sb alloy particles, resulting in nanometer-sized particles (average diameter = 16.24 nm for Bi_1_Sb_1_@C samples). In addition, the carbon skeleton can provide the space for volume expansion and also act as a three-dimensional conductive network to accelerate electron transport kinetics. Benefiting from the synergistic influence of carbon-coated and alloying effects, the Bi_1_Sb_1_@C electrodes deliver an impressive specific capacity of 167.2 mAh g^−1^ at 1 A g^−1^ over 8000 cycles. The in situ X-ray diffraction (XRD) and online differential electrochemical mass spectrometry (DEMS) were also performed to unveil the phase transformation and gases release of the Bi_1_Sb_1_@C electrode during the operation processes, respectively.

## 2. Experimental Section

### 2.1. Materials Synthesis

Bismuth (III) citrate (98%) and potassium antimony tartrate (99.5%) were purchased from Shanghai Titan Scientific Co., Ltd., Shanghai, China. All reagents were not further purified. A series of carbon-coated Bi−Sb composites were produced by a simple pyrolysis method. Bismuth (III) citrate and potassium antimony tartrate with a specific Bi/Sb ratio (3:1, 1:1, 2:3, and 1:3) were mixed evenly, then calcined at 650 °C for 1 h in an Ar flux of 20 mL min^−1^. Finally, the obtained products were washed 6 times with deionized water and dried at 80 °C. According to the Bi/Sb atomic ratios, we labeled the Bi−Sb composites as Bi_3_Sb_1_@C, Bi_1_Sb_1_@C, Bi_2_Sb_3_@C, and Bi_1_Sb_3_@C. In addition, single metal samples of Bi and Sb were also prepared in a similar process, labeled as Bi@C and Sb@C, respectively.

### 2.2. Materials Characterization

The phase compositions of Bi−Sb@C composites were characterized by an XD-3 X-ray polycrystalline powder diffractometer with Cu Kα radiation. Furthermore, Raman spectra were recorded with a 785 nm laser source in Horiba Lab HR Evolution. Scanning electron microscopy (SEM) images were observed using JSM-7610F and transmission electron microscopy (TEM) was executed using Talos F200x. The Brunauer–Emmett–Teller (BET) surface and pore distribution of the samples were calculated by N_2_ adsorption/desorption data recorded on a Gold APP V-Sorb 2800P. X-ray photoelectron spectroscopy (XPS) was applied on ESCALAB 250 XI equipped with an Al source.

### 2.3. Electrochemical Measurement

The electrochemical tests of the samples used CR2032-type cells which were assembled in a glove box (water < 0.1 ppm, oxygen < 0.1 ppm). Sodium chips were counter/reference electrodes in half cells. Whatman GF/D was chosen as a separator. The active material, carboxymethyl cellulose (CMC), and the conductive agent were uniformly mixed with water as a solvent. Copper foil acted as a current collector and was punched into 12 mm pieces as a working electrode with the active materials mass of ∼1 mg cm^−2^. As electrolytes, 1 M NaClO_4_ served as sodium salt and propylene carbonate (PC) and 5 wt.% fluoroethylene carbonate (FEC) were solvents. The galvanostatic discharge-charge tests were conducted on a battery test system (LAND CT2001A) with a voltage window from 0.01 to 2.0 V vs. Na^+^/Na. Electrochemical impedance spectroscopy (EIS) analysis was obtained by a Zahner Zennium potentiostat with a 5 mV amplitude and a frequency range of 10^−2^–10^5^ Hz. The in situ XRD data were collected using a self-made electrochemical cell device with a beryllium window (diameter is 10 mm) on one side. In order to ensure the signal strength, the XRD signal was collected using “time step” mode, where a datapoint was recorded every 0.1 degree staying for 15 s. DEMS was carried out using HPR-20EGA mass spectrometer (Hiden Analytical Ltd., Warrington, UK) combined with a Swagelok cell. High-purity Ar flow of 1 mL min^−1^ was used as a carrier gas.

## 3. Results and Discussion

A series of bimetallic Bi_x_Sb_y_ (x:y = 3:1; 1:1; 2:3; 1:3) composites embedded in carbon frameworks were synthesized by a facile pyrolysis method. Commercial potassium antimony tartrate and bismuth citrate were used as precursors. When sintering at 650 °C, the precursor decomposed with the release of a large amount of gas to form a porous carbon framework in situ. Simultaneously, the metals Bi and Sb were alloyed and trapped inside the porous carbon matrix, which prevented the aggregation and growth of alloy particles, resulting in fine alloy nanoparticles. This synthesis method can be easily scaled up for production. As shown in Appendix A, a high mass (4.5 g) of Bi_1_Sb_1_@C samples can be obtained from 10.0 g of precursor. For comparison, Bi@C and Sb@C were also synthesized by the same procedures.

Figure 1a–f and Appendix A show the SEM images of Bi@C, Bi_3_Sb_1_@C, Bi_1_Sb_1_@C, Bi_2_Sb_3_@C, Bi_1_Sb_3_@C, and Sb@C. With the increase in the Sb content, the morphology of the Bi−Sb@C composites changes from nanosheets to irregular micron-sized particles. Furthermore, the EDX results of these Bi−Sb@C composites (Appendix A) illustrate that the carbon content is between 4.02–7.44 wt.% and the actual Bi/Sb molar ratios are consistent with the designed compositions (Bi:Sb = 3:1, 1:1, 2:3 and 1:3), indicating that Bi−Sb@C composite composition is conveniently adjusted by changing the feed ratio of the precursor. To reveal the detailed microstructure and crystal structure of the Bi−Sb@C composites, the TEM measurement was performed. As shown in Figure 1g,h, the TEM images of Bi_1_Sb_1_@C demonstrate that nanosized Bi_1_Sb_1_ alloy particles are uniformly distributed and separated by amorphous carbon substrate, which is beneficial to impede the agglomeration of alloy particles during the operation of cells. The average particle size of a Bi_1_Sb_1_ alloy is approximately 16.24 ± 4.21 nm (Appendix A). Figure 1i presents three polycrystalline diffraction rings in the selected-area electron diffraction (SAED) pattern of Bi_1_Sb_1_@C, which can be indexed to the (012), (104), and (110) planes of the rhombohedral Bi_1_Sb_1_ alloy.

The phase compositions and crystal structures of the obtained samples were characterized by XRD. All XRD peaks in Figure 2a can be gratifyingly ascribed to a rhombohedral phase with $R\bar{3}m$ (166), and no carbon peaks were detected, implying the presence of amorphous carbon. The diffraction peaks of Bi_x_Sb_y_@C (x:y = 3:1; 1:1; 2:3; 1:3) composites are located between Sb (JCPDS # 35-0732) and Bi (JCPDS # 44-1246) phases. The partially enlarged view in Figure 2a exhibits that the diffraction peaks of the crystal plane (012) gradually shift to high angles with the increase in Sb content, as the Sb atoms with smaller atomic radius replaced the Bi atoms to form a solid solution with a uniform composition [17]. The Raman spectra of those samples in Figure 2b show two typical peaks in carbon materials at approximately 1330 (D-band) and 1577 cm^−1^ (G-band). The D- and G-bands represent the defective and graphitic *sp*^2^ hybridized carbon, respectively [18]. According to previous research, the D- and G-bands of amorphous carbon are located in the broad spectral range of 1320–1360 cm^−1^ and 1520–1600 cm^−1^, respectively [19]. The intensity ratios of D- and G-bands (*I_D_*/*I_G_*) in Bi@C, Bi_3_Sb_1_@C, Bi_1_Sb_1_@C, Bi_2_Sb_3_@C, Bi_1_Sb_3_@C, and Sb@C are calculated as 1.59, 1.82, 1.82, 1.89, 1.82, and 1.77, respectively. The high values of *I_D_*/*I_G_* ratios reflect that carbon materials are amorphous and disordered.

The surface elemental information was verified by XPS measurements. In the XPS survey spectrum, the characteristic peaks of Sb, Bi, and C elements can be observed, illustrating the presence of these elements in skin layer of the Bi_1_Sb_1_@C composite (Appendix A). The fitting of the Sb 3d profile for Bi_1_Sb_1_@C is shown in Figure 2c. The main peaks at 540.4 (Sb 3d_3/2_) and 531.0 eV (Sb 3d_5/2_) can be ascribed to the Sb-O bond (Sb_2_O_3_) generated by the surface oxidation process [20]. It is noteworthy that no peak of metallic Sb was detected, demonstrating that the surface of Bi_1_Sb_1_@C was completely oxidized. Furthermore, the binding energy at 532.2 eV is indexed to the typical C-O band, indicating that oxygen atoms are doped in the carbon matrix [21]. Oxygen-doped carbon materials can create sufficient Na storage sites and accelerate Na^+^ transportation [22,23]. The high-resolution spectrum of Bi 4f (Figure 2d) can be deconvoluted into two spin−orbit doublets. The peaks at 164.9 and 159.6 eV are related to Bi 4f_5/2_ and Bi 4f_7/2_ of the Bi-O bond (Bi_2_O_3_). Another two weaker peaks at 162.7 and 157.4 eV are ascribed to metallic Bi 4f_5/2_ and Bi 4f_7/2_, confirming that Bi mainly exists in the oxidized state on the surface of Bi_1_Sb_1_@C [5,24]. As displayed in Figure 2e and Appendix A, the N_2_ adsorption–desorption isotherm curves exhibit the Brunnauer–Emmett–Teller (BET) apparent surface area of 43.6, 48.6, 59.6, 34.9, 43.7, and 7.3 m^2^ g^−1^ for Bi@C, Bi_3_Sb_1_@C, Bi_1_Sb_1_@C, Bi_2_Sb_3_@C, Bi_1_Sb_3_@C, and Sb@C, respectively, which is conducive to enlarging the interface between the electrolyte and electrode and enhancing of the ions/electrons transport rate [25,26]. Figure 2f and Appendix A show that the pore-size distribution of these samples is mainly concentrated around 4 and 300 nm, which allows infiltration of sodium ions and electrolytes.

To verify the influence of alloy composition on the sodium storage performance, these samples were evaluated by CR2032-type half cells. Figure 3a–c presents the initial three cycles of CV curves of Bi@C, Bi_1_Sb_1_@C, and Sb@C at a scan rate of 0.1 mV s^−1^ with a voltage window of 0.01–2 V vs. Na^+^/Na. The reduction peaks of the first cycle differ from those of the second and third cycles, suggesting the occurrence of an activation process. In the first discharging, the irreversible cathodic peak at approximately 0.76 V vs. Na^+^/Na can be observed, which is attributed to gas release (H_2_ and CO_2_, discussed in subsequent section) and the solid electrolyte interphase (SEI) layer formation [5,10]. Note that only one dominating cathodic peak at approximately 0.3 V vs. Na^+^/Na can be detected, which reflects the multi-step transformation process with Na to form Na_3_Bi for Bi@C, Na_3_(Bi,Sb) for BiSb@C, and Na_3_Sb for Sb@C. This can be attributed to the large charge transfer resistance of the first cycle which causes the adjacent peaks to overlap with each other. In the subsequent cathodic scans, the reduction peaks at 0.66/0.56/0.47/0.18 V vs. Na^+^/Na for Bi@C, 0.60/0.45/0.31/0.22 V vs. Na^+^/Na for Bi_1_Sb_1_@C, and 0.66/0.52/0.34 V vs. Na^+^/Na for Sb@C can be observed, which are related to the stepwise alloying process [27,28]. During the anodic scans, the sharp peaks are visible at approximately 0.70/0.78, 0.75/0.84, and 0.87 V vs. Na^+^/Na for Bi@C, Bi_1_Sb_1_@C, and Sb@C, corresponding to the reversible dealloying process [29]. It can be observed that the oxidation voltages of Bi_1_Sb_1_@C are located between those of Bi@C and Sb@C, indicating that alloy composition will affect the working voltage platforms of the electrode. More importantly, the redox peaks of Bi_1_Sb_1_@C are similar to those of Bi@C, and not a combination of the step-by-step sodiation processes of Bi and Sb, implying that the Bi atom and Sb atom in Bi_1_Sb_1_@C composite may follow a synergistic sodium storage mechanism. The CV curves of the second and third cycles almost overlap, demonstrating the excellent repeatability of these materials. The CV curves of Bi_2_Sb_3_@C and Bi_1_Sb_3_@C in Appendix A also exhibit similar redox peaks to those of Bi_1_Sb_1_@C.

Figure 3d–f exhibits the initial three galvanostatic charge/discharge (GCD) profiles of Bi@C, Bi_1_Sb_1_@C, and Sb@C at 0.2 A g^−1^. A clear difference between the GCD curves of the first cycle and the following cycle can be observed, which is related to the irreversible adverse reactions (such as electrolyte decomposition, gas release, etc.) and the activation process of active materials in the first cycle. The GCD of Bi@C, Bi_1_Sb_1_@C, and Sb@C exhibit multiple potential plateaus corresponding to the CV results. The high discharge/charge capacities of 962.4/546.8 for Bi@C, 576.6/395.3 for Bi_1_Sb_1_@C, and 911.6/572.8 mAh g^−1^ for Sb@C electrodes can be obtained in the first cycle. The Bi@C, Bi_1_Sb_1_@C, and Sb@C electrode deliver initial Coulombic efficiencies (ICE) of 56.8%, 68.6%, and 62.8%, respectively. The large capacity loss in the first cycle stems from the electrolyte decomposition, SEI formation, and the trapping of sodium ions from the oxygen-doping defects of the carbon matrix [30]. Appendix A exhibits the first cycle discharge/charge capacities of 609.0/399.6 mAh g^−1^ for Bi_3_Sb_1_@C, 678.2/470.2 mAh g^−1^ for Bi_2_Sb_3_@C, and 782.0/552.3 mAh g^−1^ for Bi_1_Sb_3_@C with ICE of 65.6%, 69.3%, and 70.6%, respectively. Interestingly, the Bi−Sb@C electrode achieves a higher ICE compared to Bi@C and Sb@C electrodes, demonstrating higher reversibility of the Bi−Sb@C materials.

To estimate the alloying effect of Bi−Sb@C on cycling durability, GCD tests of Bi@C, Bi_3_Sb_1_@C, Bi_1_Sb_1_@C, Bi_2_Sb_3_@C, Bi_1_Sb_3_@C, and Sb@C electrodes were implemented at 0.2 A g^−1^ with sodium chips as the counter/reference electrode (Figure 4a). It is evident that the discharge specific capacities of these six samples during the initial dozens of cycles exhibit high composition dependence. The discharge-specific capacity is proportional to the Sb content due to the highly specific capacity of Sb. Among these electrodes with different compositions, the Bi_1_Sb_1_@C electrode achieves a satisfactory cycling durability up to 500 cycles with a reversible capacity of 201.9 mAh g^−1^. The average coulombic efficiency of the Bi_1_Sb_1_@C electrode is approximately 99.2%, implying high sodium storage reversibility. Even at a high rate of 1 A g^−1^, the ultralong cycling behaviors can be achieved for the Bi_1_Sb_1_@C electrode. The Bi_1_Sb_1_@C electrode delivers an outstanding reversible capacity of 167.2 mAh g^−1^ with a decline of only 0.0073% per cycle after 8000 cycles, demonstrating superior sodium storage reversibility (Figure 4b). In comparison, the Bi@C and Sb@C electrodes present a terrible reversible capacity.

Figure 4c illustrates the rate capability of the Bi_1_Sb_1_@C, delivering the capacities of 476.2, 416.3, 376.9, 329.3, 268.1, and 177.8 mAh g^−1^ at the current densities of 0.2–10 A g^−1^. When the test condition is reset to 0.2 A g^−1^, the capacity is restored to 380.3 mAh g^−1^, demonstrating robust structural stability of the Bi_1_Sb_1_@C. In addition, the corresponding discharge/charge plots of the Bi_1_Sb_1_@C still maintain their basic shape with a small voltage polarization compared with those of other Bi−Sb composite electrodes (Figure 4d and Appendix A). The superior sodium storage performance (e.g., long-term cycling durability and excellent rate capability) of the Bi_1_Sb_1_@C can be ascribed to the synergistic effect of the optimized alloy composition and porous carbon framework, which can alleviate the volume stress caused by volume changes, enhance the ions/electrons transfer kinetics, and prevent the aggregation of alloys during repeated cycling.

To elucidate the underlying reasons for the excellent Na storage performance of the Bi_1_Sb_1_@C electrode, the EIS test was conducted to investigate the charge transfer kinetics. Appendix A presents the Nyquist results of the Bi_1_Sb_1_@C and Sb@C electrodes for the fresh cell and charged state after the first cycle. The Nyquist results contain two parts: a depressed semicircle corresponding to the interfacial resistance (R_ct+sei_) (e.g., charge transfer resistance and SEI layer resistance) and an inclined line indicating the solid-state diffusion kinetics of Na^+^ (Warburg impedance) [31]. The circuit diagram and corresponding fitting results for the Bi_1_Sb_1_@C and Sb@C electrodes are displayed in Appendix A. For the fresh cells, the Bi_1_Sb_1_@C electrode exhibits a smaller R_ct+sei_ (98 Ω) than that (521 Ω) of the Sb@C electrode, demonstrating the rapid charge transfer kinetics due to the alloying effect. After the first cycle, the R_ct+sei_ values of Bi_1_Sb_1_@C (46 Ω) and Sb@C (52 Ω) electrodes significantly decrease, demonstrating that the formation of the SEI layer in the first cycle can accelerate the charge transfer. These results confirm the rapid charge transfer capability and stable SEI layer of the Bi_1_Sb_1_@C electrode.

To demonstrate the excellent rate capability of the Bi_1_Sb_1_@C electrode, the diffusion kinetics of Na^+^ were investigated by CV measurements with a scan rate increasing from 0.1 to 0.8 mV s^−1^. As shown in Figure 5a, the CV curves at various scan rates retain a basic shape, while the intensity of the peak current is significantly enhanced and the position of the peak current is slightly shifted as the scan rate rises, demonstrating the rapid response capability of Bi_1_Sb_1_@C electrodes at a high scan rate [31]. The diffusion rates of Na^+^ were obtained using the Randles–Sevcik equation (Equation (1)), which exhibits the linear relationship between the peak current (*i_p_*) and square root of the scan rate (*v*^0.5^) [32,33].
(1)$$i_{p}=2.69\times 10^{5}n^{3/2}AD_{Na-cv}^{0.5}v^{0.5}C$$
where *n* represents the number of transferred electrons during the sodiation/desodiation processes, A refers the surface area of the electrode, and C is defined as the Na^+^ concentration in the electrolyte. The absolute values of the fitting slope of the *i_p_* − *v*^0.5^ are positively correlated with the diffusion kinetics of Na^+^ (*D_Na-cv_*). As plotted in Figure 5b, the absolute values of the slopes of peak A1 and peak A2 (0.118 and 0.102) are significantly higher than those of peak C1, peak C2, and peak C3 (−0.047, −0.077 and −0.052), illustrating the higher diffusion kinetics of Na^+^ in the desodiation process.

The galvanostatic intermittent titration technique (GITT) technique was implemented to unveil the variation in Na^+^ (*D_Na-gitt_*) diffusion rates during the alloying/dealloying processes. The GITT curves of these six samples in the first cycle are displayed in Figure 5c and Appendix A. GITT tests were performed with a pulse current of 0.05 A g^−1^. The diffusion kinetics of Na^+^ are estimated by the following equation (Equation (2)) [34,35]:(2)$$D_{Na-gitt}=\frac{4}{\pi \tau }{\left(\frac{n_{m}V_{m}}{A}\right)}^{2}{\left(\frac{\Delta E_{s}}{\Delta E_{\tau }}\right)}^{2}$$
where *τ* stands for the time duration of the current pulse; *n_m_*, *V_m_*, and *A* represent the molar mass/volume of the electrode materials and the surface area of electrodes, respectively; and ∆Es and ∆Eτ represent the steady-state voltage change and transient voltage change, respectively (Appendix A). The values of ∆Es and ∆Eτ are obtained from GITT curves in Figure 5c and Appendix A. As displayed in Figure 5d and Appendix A, the diffusion coefficients of Na^+^ during the charging process are higher than those during the discharging process, which is similar to the above results obtained from the CV technique. Among these samples, the Bi_1_Sb_1_@C electrode exhibits the highest Na^+^ diffusion rate, ranging from 10^−11^ to 10^−13^ cm^2^ s^−1^ during the entire charging/discharging processes, confirming its fast Na^+^ transport kinetics.

To uncover the reaction mechanism of Na in Bi−Sb@C, the in situ XRD measurement was executed to track the phase transition process of B_1_Sb_1_@C electrodes during the first cycle. The in situ cell was controlled by a galvanostatic charge–discharge test at 0.05 A g^−1^. Figure 6 presents the in situ XRD results in the form of line (Figure 6a) and contour (Figure 6b) plots. The corresponding charge–discharge curves are shown on the left side. Three diffraction peaks BeO (JCPDS # 35-0818) originating from the oxidation of the beryllium window are located at 38.5°, 41.2°, and 43.9°. The peak of stainless-steel mesh (SS-mesh) can be detected at 43.4°. The intensity and position of these peaks did not change throughout the discharge/charge processes. The final sodiated products of metals Sb and Bi are hexagonal Na_3_Sb (JCPDS # 65-3523) and Na_3_Bi (JCPDS# 65-3525) phases, which have similar crystal structures [18,36]. Therefore, we propose that Bi and Sb may follow a synergistic sodium storage process rather than a step-by-step process.

During the first discharge process (stage 1, 2.4–0.45 V vs. Na^+^/Na), the characteristic peaks of the rhombohedral Bi_1_Sb_1_ phase at 27.6° and 40.4° are gradually attenuated, indicating the occurrence of the alloying reaction of Na with Bi_1_Sb_1_ composite. Discharging to ~0.45 V vs. Na^+^/Na, the peaks of the Bi_1_Sb_1_ composite disappear, while four new peaks at 18.3°, 25.8°, 31.9°, and 37.1° could be detected, suggesting that the Bi_1_Sb_1_ composite is transformed into the Na(Bi,Sb) phase. As the discharge process proceeds (stage 2, 0.45–0.01 V vs. Na^+^/Na), the peak intensity of Na(Bi,Sb) gradually weakens and disappears, while the characteristic peaks of the Na_3_(Bi,Sb) phase appear at 18.3°, 18.7°, 20.9°, 32.9°, and 33.7° and reach the strongest intensity at 0.01 V vs. Na^+^/Na. This confirms that the intermediate phase of Na(Bi,Sb) continues the alloying reaction with Na to form the Na_3_(Bi,Sb) phase [37]. During the charging process (stage 3, 0.01–0.76 V vs. Na^+^/Na), the desodiation reaction proceeds. The Na_3_(Bi,Sb) phase reversibly transforms into the Na(Bi,Sb) phase [37]. It should be noted that the peak position of the Na(Bi,Sb) phase shifts to a lower angle, implying that its crystal structure has undergone reconstruction. When the voltage increases to 2 V vs. Na^+^/Na, the peaks of the Bi_1_Sb_1_ phase at 27.2° and 39.7° reappear and shift to a lower angle simultaneously with the diminishment and disappearance of the Na(Bi,Sb) phase. Afterwards, the reversible phase evolution process can be observed in the charging stage.

Online DEMS is a powerful tool for analyzing gaseous products during battery operation [38]. Therefore, DEMS tests were carried out to explore the gas evolution and the relationship between gas generation and the voltage of the cells during the operation processes. The voltage curves of the B_1_Sb_1_@C//Na half-cell (top panel) and the DEMS results from the initial cycle (bottom panel) are displayed in Figure 7. Before the test, the inlet of Swagelok cells was connected to the gas source and the outlet was connected to the DEMS system for two hours with the continuous Ar flow for the baseline calibration. When discharging to 0.72 V vs. Na^+^/Na, the DEMS signals at *m*/*z* = 2 (H_2_, red line) and *m*/*z* = 44 (CO_2_, pink line) began to be detected and reached the maximum at approximately 0.52 V vs. Na^+^/Na, implying that gas products were continuously generated during this process. The H_2_ stems from the side reaction of trace water in the cell, including the electrolyte, separators, and sodium foil [39,40]. The CO_2_ release originates mainly from the decomposition of eater-based electrolytes forming the SEI layer [41]. During the subsequent charging, no significant peaks of H_2_ and CO_2_ were observed, suggesting that the electrolyte decomposition did not occur during the charging process. CH_4_ (blue line, m/z = 16) and O_2_ (green line, m/z = 32) DEMS signals gradually decreased, but no obvious peak appeared throughout the cycling, indicating that these gases would not be released at low voltages (less than 2 V vs. Na^+^/Na) [38].

## 4. Conclusions

In summary, a series of bismuth−antimony composites with different compositions confined in a porous carbon matrix were successfully fabricated by a simple pyrolysis method. An optimized Bi−Sb composition, porous carbon matrix, and nano-sized particles can alleviate stress changes derived from repeated volume expansion/contraction during the charging/discharging processes. When evaluated as anodes of SIBs, composition-optimized Bi_1_Sb_1_@C electrodes exhibit long-term cycling stability, delivering 167.2 mAh g^−1^ after 8000 cycles at high rate of 1 A g^−1^. In addition, the synergistic sodium storage mechanism of Bi−Sb composites during the sodiation/desodiation processes was verified by the in situ XRD results involving a synchronous sodiation process. Our findings propose reasonable insight into the sodium storage mechanism of bimetallic alloys and provide new avenues for compositional regulation and in situ carbon coating synthesis of advanced alloying-type anodes for SIBs.

## Figures and Tables

**Figure 1 materials-16-02189-f001:**
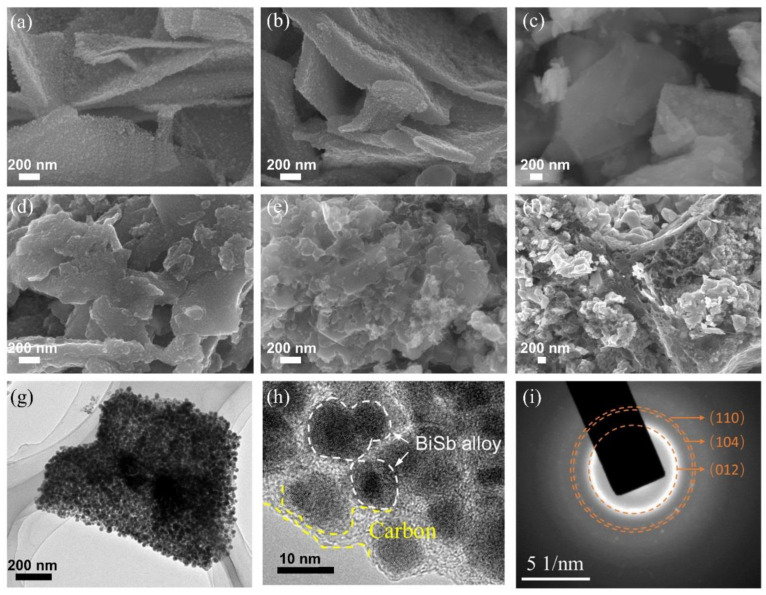
SEM images of the (**a**) Bi@C, (**b**) Bi_3_Sb_1_@C, (**c**) Bi_1_Sb_1_@C, (**d**) Bi_2_Sb_3_@C, (**e**) Bi_1_Sb_3_@C, and (**f**) Sb@C samples. (**g**) TEM image, (**h**) HRTEM image, and (**i**) SAED pattern of Bi_1_Sb_1_@C.

**Figure 2 materials-16-02189-f002:**
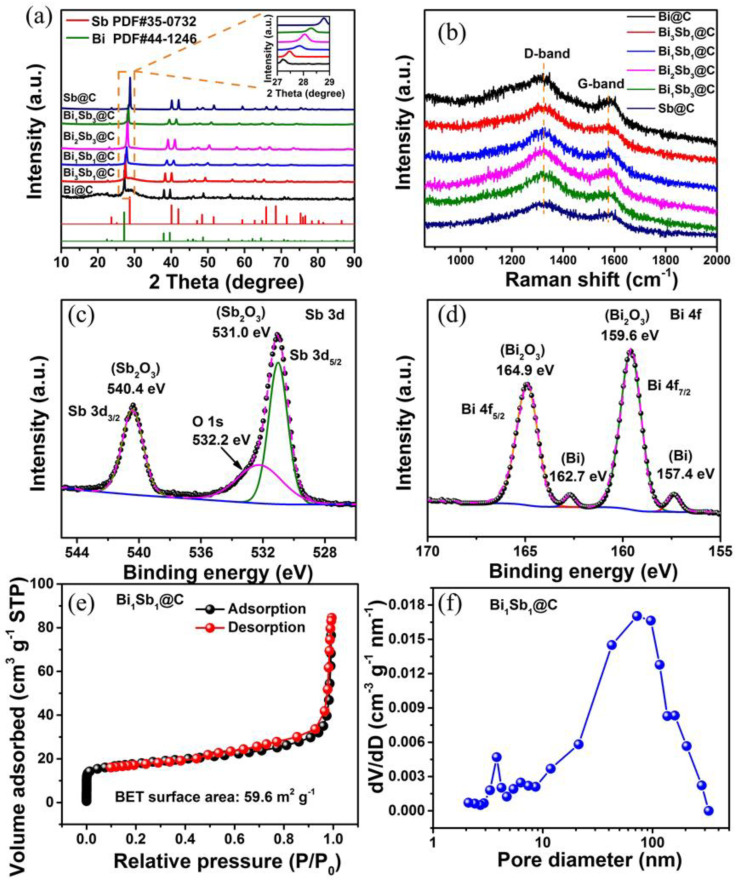
(**a**) XRD patterns in the range of 10°–90° and (**b**) Raman results of the Bi−Sb@C samples, respectively. High-resolution XPS spectra of (**c**) Sb 3d and (**d**) Bi 4f for the Bi_1_Sb_1_@C sample. (**e**) N_2_ adsorption/desorption isotherms and (**f**) pore size distribution curves of the Bi_1_Sb_1_@C sample.

**Figure 3 materials-16-02189-f003:**
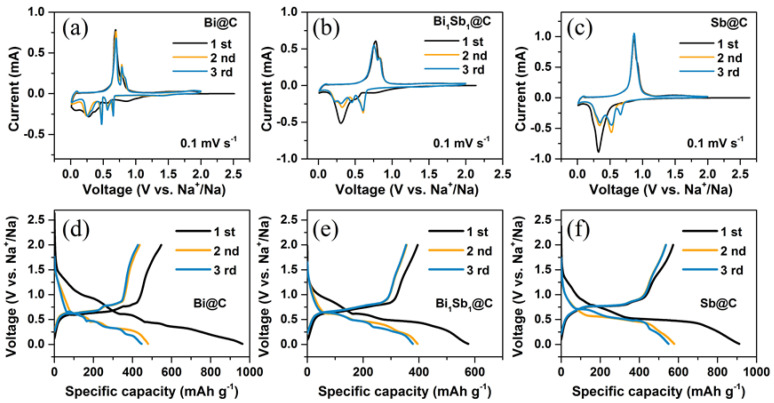
(**a**–**c**) Initial three cycles of CV curves and (**d**–**f**) the voltage curves of Bi@C, Bi_1_Sb_1_@C, and Sb@C samples during the initial three cycles at 0.2 A g^−1^.

**Figure 4 materials-16-02189-f004:**
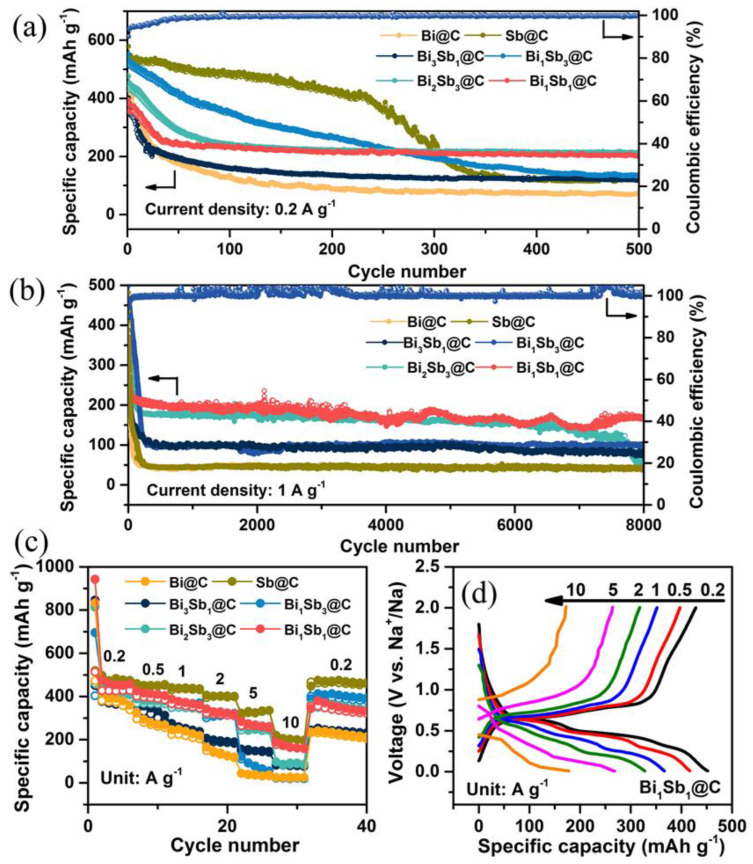
Na storage performance of the Bi−Sb@C samples. Cycling stability of the Bi−Sb@C samples electrodes at current densities of (**a**) 0.2 A g^−1^ and (**b**) 1 A g^−1^. (**c**,**d**) Rate capability and discharge–charge profiles of Bi−Sb@C electrodes at various current densities from 0.2 to 10 A g^−1^.

**Figure 5 materials-16-02189-f005:**
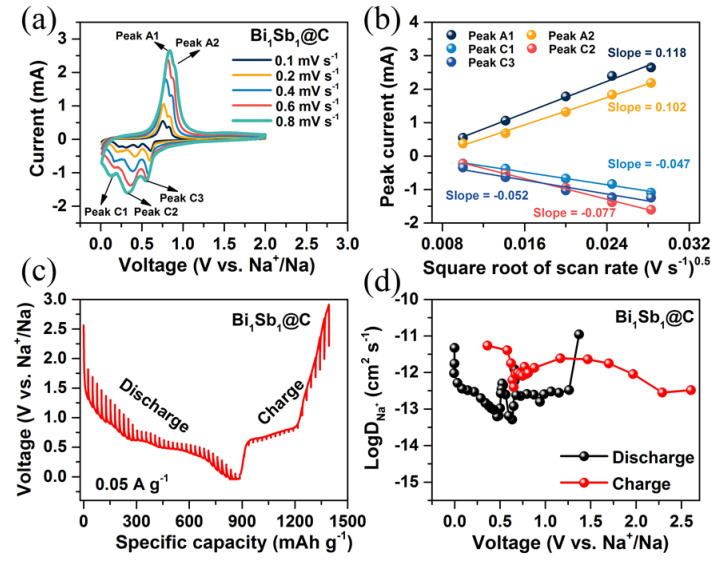
(**a**) CV results of the Bi_1_Sb_1_@C electrode at sweep rates from 0.1 to 0.8 mV s^−1^. (**b**) Linear relationship between *i_p_* and *v*^0.5^. (**c**) GITT curves of the Bi_1_Sb_1_@C sample at 0.05 A g^−1^. (**d**) Corresponding diffusion rate for Na^+^ calculated from the GITT results during the first cycle.

**Figure 6 materials-16-02189-f006:**
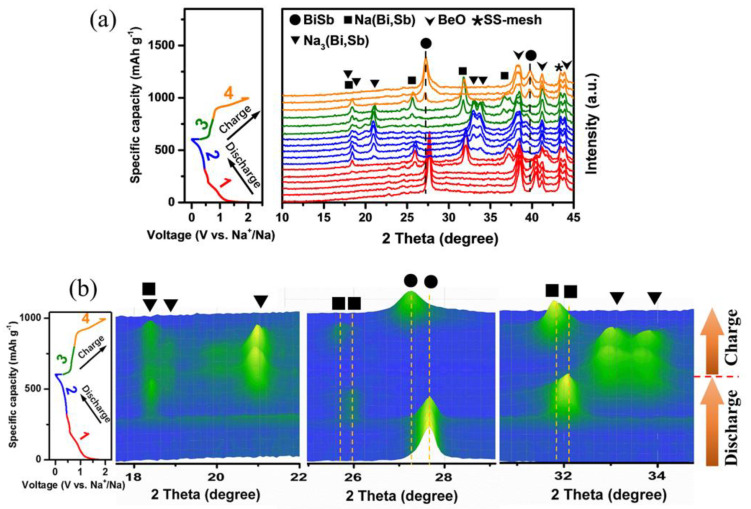
The (**a**) line plot and (**b**) contour plot for the in situ XRD patterns of the Bi_1_Sb_1_@C electrode obtained from the first cycle. The corresponding discharge/charge curves are tested at 0.05 A g^−1^.

**Figure 7 materials-16-02189-f007:**
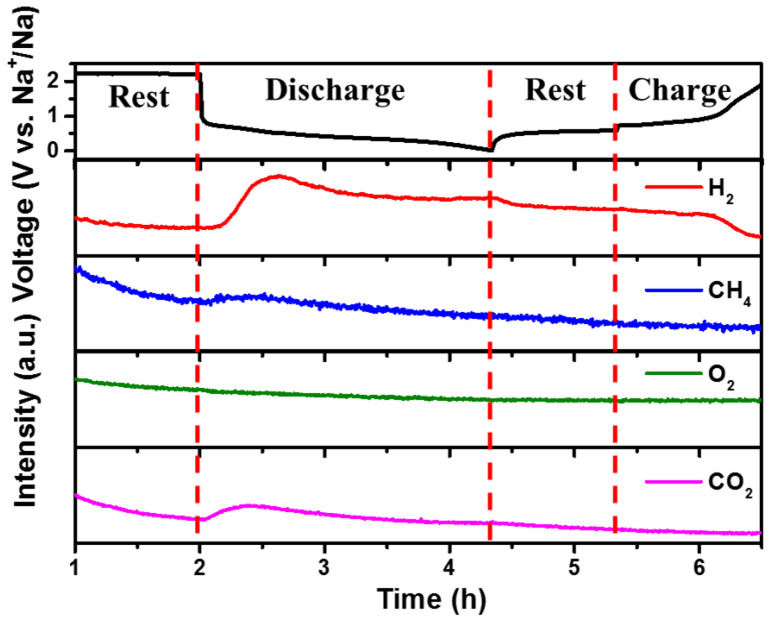
The on-line DEMS results collected by a Swagelok cell with Bi_1_Sb_1_@C electrode and PC, 5 wt.% FEC, 1 M NaClO_4_ electrolyte during rest and initial discharge–charge processes.

## Data Availability

All data are available within the paper and its Appendix A or from the corresponding authors upon request.

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
