# Peer review of "Bismuth−Antimony Alloy Embedded in Carbon Matrix for Ultra-Stable Sodium Storage"

_materials, 2023, doi:10.3390/ma16062189_

Round 1

Reviewer 1 Report

Review on

BISMUTH−ANTIMONY ALLOY EMBEDDED IN CARBON MATRIX FOR ULTRA-STABLE SODIUM STORAGE

The authors studied a new material for sodium-ion batteries, namely an  anode material, a bimetallic Bi−Sb alloy embedded in a porous carbon matrix, with a  pyrolysis method. Comparative results are presented for Bi@C, BixSby@C and Sb@C.

 The anode material has been investigated with spectroscopy and surface analysis method as XRD, Sem, TEM, Raman, XPS and also electrochemical method as CV, charge-discharge curves, cycle and rate performance, electrochemical impedance spectroscopy. The best obtained alloy is the optimized Bi1Sb1@C electrode which  exhibits an excellent electrochemical performance with ultralong cycle life (167.2 mAh g−1 at 1 A g−1 over 8000 cycles).

Just an observation- personally I dislike the word “operando” both in introduction and key words. It will be better if you can find another word/expression. For Key words, it is enough to write XRD.

Excepting the minor observation above, I consider that article could be published in this form.

Reviewer 2 Report

Overview:

In their study (materials-2256739), authors report on the BixSby nanoparticles encapsulated in amorphous carbon, variation of their structure with precursor content change and their properties as the electrode of Na-ion battery. The study is of interest, as several works have recently been dedicated to the electrochemical application of Sb-Bi [10.1016/j.jpowsour.2021.230826], Bi@C [10.1016/j.electacta.2022.141405] (for Li storage), Bi [10.1016/j.mtphys.2021.100486]. Studies are rigorous and well-designed, and the manuscript is mostly well-written. However, several (mostly minor) issues should be resolved prior to the production of the paper.

Major comments:

1) Carbon is known to be a plausible electrode for Na storage [Komaba et al., Adv.Funct.Mater. 21, 20, 2011. 10.1002/adfm.201100854]. However, authors report on the sodiation/desodiation of the BixSby component only. This emerges the question: does carbon concentration vary for different samples, and what is the carbon component contribution to the electrochemical properties of the samples? It would be beneficial to discuss this aspect by analyzing the elemental composition of the samples assessed by survey XPS and EDX.

2) As authors report on the annealing of BiC6H5O7 and K2Sb2(C4H2O6)2 precursors, the discussion related to the purity of the material should be provided: isn’t resulting amorphous carbon remain doped with O or K? How does it affect the properties of the material?

Minor comments:

3) Page 2, “Wang et al. reported that the CoSb nanocrystalline encapsulated”: did you mean “nanocrystallites” or “nanocrystalline phase”?

4) Page 2, what did you mean by “Based on the infinite solid solution”?

5) Page 2, “then placed in a tube furnace at 650 °C for 1 h in an Ar atmosphere.”:  what were the Ar pressure/flux?

6) Page 2, “Bismuth (III) citrate and potassium antimony tartrate with a specific stoichiometric ratio were mixed evenly”: did you mean that the stoichiometry of BiC6H5O7 and K2Sb2(C4H2O6)2 compounds somehow changed? Or that the variation of the ratio of these components lead to the variation of the stoichiometry of the structure? Please revise the fragment.

7) “Materials Characterization” section: how were the materials prepared for TEM (separated from bulk sample, placed on the TEM grids)?

8) Figure S2(d,f,h,j): in the insets, do you present the result of the EDX studies or the Bi/Sb ratio assessed via EDX? According to EDX studies, is there carbon in the materials?

9) Fig. 1: in (g) subfigure, (g) notation, scale caption and scale bar are barely visible; in (h), scale notation, “carbon” caption and scale bar are barely visible: please create a black outline for these objects.

10) Page 4: although XPS is discussed in detail, for O1s line nothing is stated except its position. Isn’t O1s positioning at 532.2 eV typical for C-O [Applied Physics A (2022) 128:929, https://doi.org/10.1007/s00339-022-06062-2], thus indicating that oxidized carbon is prevailing at subsurface layers? It is quite common for a-C/ta-C structures.

11) Page 9, could you explain the physical meaning of ∆Es and ∆Eτ values and how they are derived from GITT curves?

12) In Fig. 6b, is it possible to mark the beginning of charge and discharge processes?

13) Is there any physical meaning in the low initial intensity of the CH4- and O2-related curves in Fig. 7? Current layout leaves empty space in these subfigures.

14) In Fig. 7, authors state that “DEMS signals of CH4… gases remained stable throughout the cycling”; however, CH4-related signal seems to be decreased from 0.5×max_value to 0.2×max_value throughout the cycle. Could you comment on that?

15) In page 14, by “writing the review and editing”, you probably meant “writing – review and editing”, as this manuscript is not a review.

Round 2

Reviewer 2 Report

I appreciate the authors’ feedback, elaborate comments and rigorous revision. I recommend the manuscript is subjected to some “fine tuning” prior to its production.

1) Section “Electrochemical measurement”: by “frequency range of 100 kHz - 10 mHz”, did you mean “MHz”?

2) In Section “3. Results and discussion”: “Typically, commercial potassium antimony tartrate and bismuth citrate were used as precursors”. “Typically” should be removed, as you currently refer to your experiments.

3) In page 6, why do you refer to Bi1Sb1@C as to the “Bi1Sb1@C alloy”? It is a composite material rather than an alloy. In page 4, “Bi−Sb@C alloy(s)” are also mentioned: please revise the wording throughout the text.

4) Page 6, by “indicating that C-O is dominant in the carbon matrix”, did you mean that “oxygen is predominantly bonded with carbon”? C-O is not predominant, as EDX indicates that O amount is undetectable.

5) In page 7, by “Figure 3a−c presents the initial three cycles CV curves”, did you mean “three cycles of CV curves”?

6) Authors correctly indicate that the lines observed in the studied Raman spectra are D and G lines. However, their positions and intensity ratio should be compared with literature, as the C=C-vibrations-related line can also be indicative of graphite, if located at 1580-1600 cm-1, or polyenes (its position may shift from 15xx to 1680 cm-1 for short sp2-hybridized polymeric chains). Consider comparing your results with the ones reported in [Streletskiy et al. “Amorphous Carbon Films with Embedded Well-Dispersed Nanodiamonds: Plasmon-Enhanced Analysis and Possible Antimicrobial Applications”. Magnetochemistry. Volume 8, Issue 12. 10.3390/magnetochemistry8120171], where the amorphous-carbon-related intervals of 1320–1360 cm−1 and 1520–1600 cm−1 were reported for D and G-line positions and ID/IG varied in the range of 0.6-2.6.

7) Could you please discuss what does the “IR drop” means in Fig. S13?

8) I cannot see the changes between previous and revised versions of Fig. 6, which should have been revised in accordance to the previous minor comment 12:

12) In Fig. 6b, is it possible to mark the beginning of charge and discharge processes?”

9) In conclusions, page 18, by “a synchronous sodiated process” did you mean “sodiation process”?
